

# Development of an RNA interference (RNAi) gene knockdown protocol in the anaerobic gut fungus *Pecoramyces ruminantium* strain C1A

Shelby S. Calkins[1], Nicole C. Elledge[1,4], Katherine E. Mueller[1], Stephen M. Marek[2], MB Couger[3], Mostafa S. Elshahed[1] and Noha H. Youssef[1]

[1] Department of Microbiology and Molecular Genetics, Oklahoma State University, Stillwater, OK, USA
[2] Department of Entomology and Plant Pathology, Oklahoma State University, Stillwater, OK, USA
[3] High Performance Computing Center, Oklahoma State University, Stillwater, OK, USA
[4] Current affiliation: University of Texas A&M Corpus Christi, Department of Life Sciences, Marine Biology Program, USA

Corresponding author
Noha H. Youssef, noha@okstate.edu

## ABSTRACT

Members of the anaerobic gut fungi (AGF) reside in rumen, hindgut, and feces of ruminant and non-ruminant herbivorous mammals and reptilian herbivores. No protocols for gene insertion, deletion, silencing, or mutation are currently available for the AGF, rendering gene-targeted molecular biological manipulations unfeasible. Here, we developed and optimized an RNA interference (RNAi)-based protocol for targeted gene silencing in the anaerobic gut fungus *Pecoramyces ruminantium* strain C1A. Analysis of the C1A genome identified genes encoding enzymes required for RNA silencing in fungi (Dicer, Argonaute, *Neurospora crassa* QDE-3 homolog DNA helicase, Argonaute-interacting protein, and *Neurospora crassa* QIP homolog exonuclease); and the competency of C1A germinating spores for RNA uptake was confirmed using fluorescently labeled small interfering RNAs (siRNA). Addition of chemically-synthesized siRNAs targeting D-lactate dehydrogenase (*ldhD*) gene to C1A germinating spores resulted in marked target gene silencing; as evident by significantly lower *ldhD* transcriptional levels, a marked reduction in the D-LDH specific enzymatic activity in intracellular protein extracts, and a reduction in D-lactate levels accumulating in the culture supernatant. Comparative transcriptomic analysis of untreated versus siRNA-treated cultures identified a few off-target siRNA-mediated gene silencing effects. As well, significant differential up-regulation of the gene encoding NAD-dependent 2-hydroxyacid dehydrogenase (Pfam00389) in siRNA-treated C1A cultures was observed, which could possibly compensate for loss of D-LDH as an electron sink mechanism in C1A. The results demonstrate the feasibility of RNAi in anaerobic fungi, and opens the door for gene silencing-based studies in this fungal clade.

## INTRODUCTION

The role played by non-coding RNA (ncRNA) molecules in epigenetic modulation of gene expression at the transcriptional and post-transcriptional levels is now well recognized (*Catalanotto, Cogoni & Zardo, 2016*). Small interfering RNAs (siRNA) are short (20–24 nt) double stranded RNA molecules that mediate post-transcriptional regulation of gene expression and gene silencing by binding to mRNA in a sequence-specific manner (*Quoc & Nakayashiki, 2015*). The process of RNA interference (RNAi) has been independently documented in fungi (*Chang, Zhang & Liu, 2012*; *Cogoni & Macino, 1997b*; *Romano & Macino, 1992*), animals and human cell lines (*Atayde, Tschudi & Ullu, 2011*; *Chiu & Rana, 2002*), as well as plants (*Fang & Qi, 2016*). The fungal RNAi machinery has been investigated in several model fungi, e.g., *Neurospora crassa* (*Romano & Macino, 1992*), *Mucor circinelloides* (*Nicolás, Torres-Martínez & Ruiz-Vázquez, 2003*), and *Magnaporthe oryzae* (*Kadotani et al., 2003*; *Kadotani et al., 2004*), and encompasses: (1) Dicer (Dic) enzyme(s): RNaseIII dsRNA-specific ribonucleases that cleave double stranded RNA (dsRNA) to short (20–25 bp) double stranded siRNA entities; (2) Argonaute (Ago) protein(s), the core component of the RNA-induced silencing complex (RISC) which binds to the dicer-generated siRNAs and other proteins and cleaves the target mRNA; (3) RNA-dependent RNA polymerase (RdRP) enzyme (present in the majority, but not all fungi) that aids in amplifying the silencing signal through the production of secondary double stranded siRNA molecules from single stranded mRNAs generated by the RISC complex; (4) DNA helicase, *Neurospora crassa* QDE-3 homolog (*Pickford et al., 2002*), that aids in the production of the aberrant RNA to be targeted by RdRP; and (5) Argonaute-interacting protein, *Neurospora crassa* QIP homolog (*Maiti, Lee & Liu, 2007*), an exonuclease that cleaves and removes the passenger strand from the siRNA duplex.

The phenomenon of RNA interference could induce gene silencing due to the action of endogenously produced microRNA (miRNA), or could be triggered due to the introduction of foreign siRNA (e.g., due to viral infection or genetic manipulation). Under normal physiological conditions, RNAi is thought to play a role in endogenous regulation of gene expression (*Bartel, 2004*), development of resistance to viruses (*Hammond et al., 2008a*; *Segers et al., 2007*; *Sun, Choi & Nuss, 2009*; *Zhang et al., 2008*), and silencing the expression of transposons (*Murata et al., 2007*; *Nolan et al., 2005*). On the other hand, the introduction of foreign siRNA could be utilized for targeted, sequence-specific, gene knockdown in fungi (*Quoc & Nakayashiki, 2015*; *Chang, Zhang & Liu, 2012*; *Romano & Macino, 1992*). Indeed, demonstration of the feasibility of RNAi approaches for targeted gene silencing has been shown in Ascomycota (*Romano & Macino, 1992*; *Abdel-Hadi et al., 2011*; *Barnes, Alcocer & Archer, 2008*; *Eslami et al., 2014*; *Jöchl et al., 2009*; *Kalleda, Naorem & Manchikatla, 2013*; *Li et al., 2012*; *Moazeni et al., 2012*; *Moazeni et al., 2014*; *Mousavi et al., 2015*; *Penn et al., 2015*; *Prakash, Manjrekar & Chattoo, 2016*), Basidiomycota (*Caribé dos Santos et al., 2009*; *Matityahu et al., 2008*; *Nakade et al., 2011*; *Namekawa et al., 2005*; *Skowyra & Doering, 2012*), and Mucoromycota (*Gheinani et al., 2011*; *Nicolas et al., 2008*); and RNAi-based protocols were used to infer the putative roles of several genes or simply as a proof of principle.

The anaerobic gut fungi (AGF) represent a basal fungal phylum (Neocallimastigomycota) that resides in the herbivorous gut and plays an important role in enhancing plant biomass metabolism by the host animals (*Gruninger et al., 2014*). The AGF have multiple potential biotechnological applications such as a source of lignocellulolytic enzymes (*Cheng et al., 2014*; *Kwon et al., 2016*; *Lee et al., 2015*; *Morrison, Elshahed & Youssef, 2016*; *Wang, Chen & Hseu, 2014*; *Wei et al., 2016a*; *Wei et al., 2016b*), direct utilization of AGF strains for sugar extraction from plant biomass in enzyme-free biofuel production schemes (*Ranganathan et al., 2017*), additives to biogas production reactors (*Nkemka et al., 2015*; *Procházka et al., 2012*), and feed additives for livestock (*Dey et al., 2004*; *Lee, Ha & Cheng, 2000*; *Paul et al., 2011*; *Paul et al., 2004*; *Saxena et al., 2010*; *Sehgal et al., 2008*; *Tripathi et al., 2007*). However, the strict anaerobic nature of AGF renders genetic manipulation procedures involving plating and colony selection extremely cumbersome. Consequently, there are currently no protocols for transformation, gene insertion, gene deletion, or sequence-specific homologous recombination-based genetic manipulation in AGF, hindering in-depth investigation of their biotechnological potential.

We here report on the development of an RNAi-based protocol for targeted gene knockdown in the anaerobic gut fungal isolate *Pecoramyces ruminantium* strain C1A. The protocol does not involve transformation, and does not require homologous recombination, or colony selection. We demonstrate the uptake of chemically synthesized short double stranded siRNA by germinating spores of *P. ruminantium* strain C1A, and subsequently demonstrate the feasibility of using this approach for silencing D-lactate dehydrogenase (*ldhD*) gene. We finally examine the off-target effects of *ldhD* gene knockdown, as well as the impact of inhibiting D-lactate production on the glycolytic and fermentation pathways in C1A.

## MATERIALS AND METHODS

### Microorganism and culture maintenance
*Pecoramyces ruminantium* strain C1A was isolated previously in our laboratory (*Hanafy et al., 2017*) and maintained by biweekly transfers into an antibiotic-supplemented rumen-fluid-cellobiose medium (RFC) as described previously (*Calkins et al., 2016*).

### Identification and phylogeny of RNAi complex in anaerobic fungi
The occurrence of genes encoding Dic, Ago, RdRP, QIP, and QDE3 proteins was examined in the genome of *P. ruminantium* C1A (*Youssef et al., 2013*) (Genbank accession number ASRE00000000.1), as well as in three additional publicly available Neocallimastigomycota genomes (*Solomon et al., 2016*) (Genbank accession numbers: MCOG00000000.1, MCFG00000000.1, MCFH00000000.1). The phylogeny of the translated amino acid sequences of identified homologues was compared to fungal and eukaryotic homologues in MEGA7. Representative sequences were aligned using ClustalW and the aligned sequences were manually refined and used to construct Neighbor Joining trees in Mega7 (*Kumar, Stecher & Tamura, 2016*) with bootstrap values calculated based on 100 replicates.
### RNAi experimental design
#### Choice of delivery procedure

Delivery of the inhibitory RNA molecules to fungal cultures is commonly achieved using appropriate vectors that either express short hairpin RNA (*Hammond et al., 2008b*; *Hammond & Keller, 2005*; *Nakayashiki et al., 2005*), or individual sense and antisense RNA strands that will subsequently be annealed into dsRNA (*Cogoni & Macino, 1997b*; *Patel et al., 2008*). The process involves transformation (PEG-CaCl$_2$-mediated into protoplasts, Li acetate-mediated, *Agrobacterium*-mediated, or via electroporation) and necessitates transformants' selection on marker (usually hygromycin) plates. Alternatively, direct delivery of exogenous, chemically synthesized short double stranded RNA (siRNA) has also been utilized for targeted gene silencing in fungi (*Eslami et al., 2014*; *Jöchl et al., 2009*; *Kalleda, Naorem & Manchikatla, 2013*; *Mousavi et al., 2015*; *Khatri & Rajam, 2007*). This approach exploits the machinery for nucleic acids uptake, and the natural competence of the germinating spore stage observed in the filamentous fungus *Aspergillus* (*Jöchl et al., 2009*). Due to the strict anaerobic nature of AGF which would hinder the process of transformation and selection on plates, we opted for direct addition of chemically synthesized siRNA to C1A germinating spores, in spite of its reported lower efficacy (*Kalleda, Naorem & Manchikatla, 2013*).

#### dsRNA synthesis

We targeted D-lactate dehydrogenase (*ldhD*) gene encoding D-LDH enzyme (EC 1.1.1.28). D-LDH is an NAD-dependent oxidoreductase that reduces pyruvate to D-lactate, a major fermentation end product in C1A (*Ranganathan et al., 2017*). Only a single copy of *ldhD* (996 bp in length) was identified in C1A genome (IMG accession number 2511055262 within the C1A genome: https://img.jgi.doe.gov/cgi-bin/m/main.cgi?section=TaxonDetail&page=taxonDetail&taxon_oid=2510917007). A 21-mer siRNA targeting positions 279-298 in the *ldhD* gene transcript (henceforth *ldhD*-siRNA) was designed using Dharmacon® siDesign center (http://dharmacon.gelifesciences.com/design-center/) with the sense strand being 5′-CGUUAGAGUUCCAGCCUAUUU-3′, and the antisense strand being 5′-AUAGGCUGGAACUCUAACGUU-3′. Included within the designed siRNAs were 3′ overhanging UU dinucleotides to increase the efficiency of target RNA degradation as suggested before (*Elbashir et al., 2001*). The siRNA was ordered from Dharmacon® (LaFayette, CO, USA) as 21-mer duplex (double stranded) with a central 19-bp duplex region and symmetric UU dinucleotide 3′ overhangs on each end. The 5′ end of the antisense strand was modified with a phosphate group required for siRNA activity (*Chiu & Rana, 2003*), while the 5′ end of the sense strand was modified with a Cy-3 fluorescent dye to facilitate visualization of the siRNA uptake by C1A germinating spores. In addition, a 21-mer duplex that should not anneal to any of C1A's mRNA transcripts (henceforth unrelated-siRNA) was also designed and used as a negative control with the sense strand being 5′-UCGUUGGCGUGAGCUUCCAUU-3′, and the antisense strand being 5′-UGGAAGCUCACGCCAACGAUU-3′. The unrelated-siRNA was modified in the same way as the *ldhD* siRNA.

### RNAi protocol

The basic protocol employed is shown in Fig. 1. Strain C1A was grown on RFC-agar medium in serum bottles at 39 °C in the dark as described previously (*Calkins et al., 2016*) until visible surface colonies are observed (usually 4–7 days). Surface growth was then flooded by adding 10 ml sterile anoxic water followed by incubation at 39 °C (*Calkins et al., 2016*). During this incubation period, spores are released from surface sporangia into the anoxic water. Previous work has shown that the duration of incubation with the flooding solution has a major impact on the spore developmental stage, where exclusively active flagellated spores were observed in incubations shorter than 30 min, while 90–100-minute incubation exclusively produced germinating spores. The onset of spore germination was observed at 75–80 min during incubation with the flooding solution (*Calkins et al., 2016*). Germinating spores were previously shown to be most amenable for accumulating the highest amount of exogenously added nucleic acids (*Jöchl et al., 2009*). We, therefore, reasoned that addition of chemically synthesized siRNA to the sterile anoxic flooding water at the onset of spore germination (at around 75 min from the onset of flooding) followed by re-incubation at 39 °C for 15 additional minutes (for a total of 90-minute incubation period) would allow for uptake of the siRNA by the germinating spores. Chemically synthesized siRNA was added from a stock solution constituted in a sterile anoxic RNase-free siRNA buffer (60 mM KCl, 6 mM HEPES-pH 7.5, and 0.2 mM $MgCl_2$) to the desired final concentration. Initial experiments were conducted using Cy3-labeled *ldhD*-siRNA molecules to test the uptake of siRNA by the germinating spores. Subsequent experiments were conducted using unlabeled siRNA. Following siRNA addition and incubation, spores were gently recovered from the serum bottle using a 16G needle and used to inoculate fresh RFC media bottles (*CalFkins et al., 2016*), and the impact of silencing *ldhD* gene on gene expression, enzyme activities, and D-lactate concentrations was assessed in these cultures. Controls included treatments with unrelated-siRNA, as well as cultures with no siRNA addition.

## Impact of *ldhD* gene knockdown on *ldhD* transcriptional levels, D- LDH enzyme activity, and D-lactate production in strain C1A

The supernatant of both siRNA-treated and control C1A cultures was periodically sampled (0.5 ml) and used for D-lactate quantification. The amount of fungal biomass at the time of quantification was derived from the headspace pressure as previously described (*Ranganathan et al., 2017*). The fungal biomass was vacuum filtered on 0.45 $\mu$m filters, and immediately crushed in a bath of liquid nitrogen using a mortar and pestle as described previously (*Calkins & Youssef, 2016*). The crushed cells were then poured into 2 separate 15-mL plastic falcon tubes, and stored at −80 °C for subsequent RNA, and protein extraction, respectively.

### D-Lactate quantification

D-lactate was determined in the culture supernatant using the D-Lactate Assay Kit (BioAssay Systems, Hayward, CA, USA) following the manufacturer's instructions.
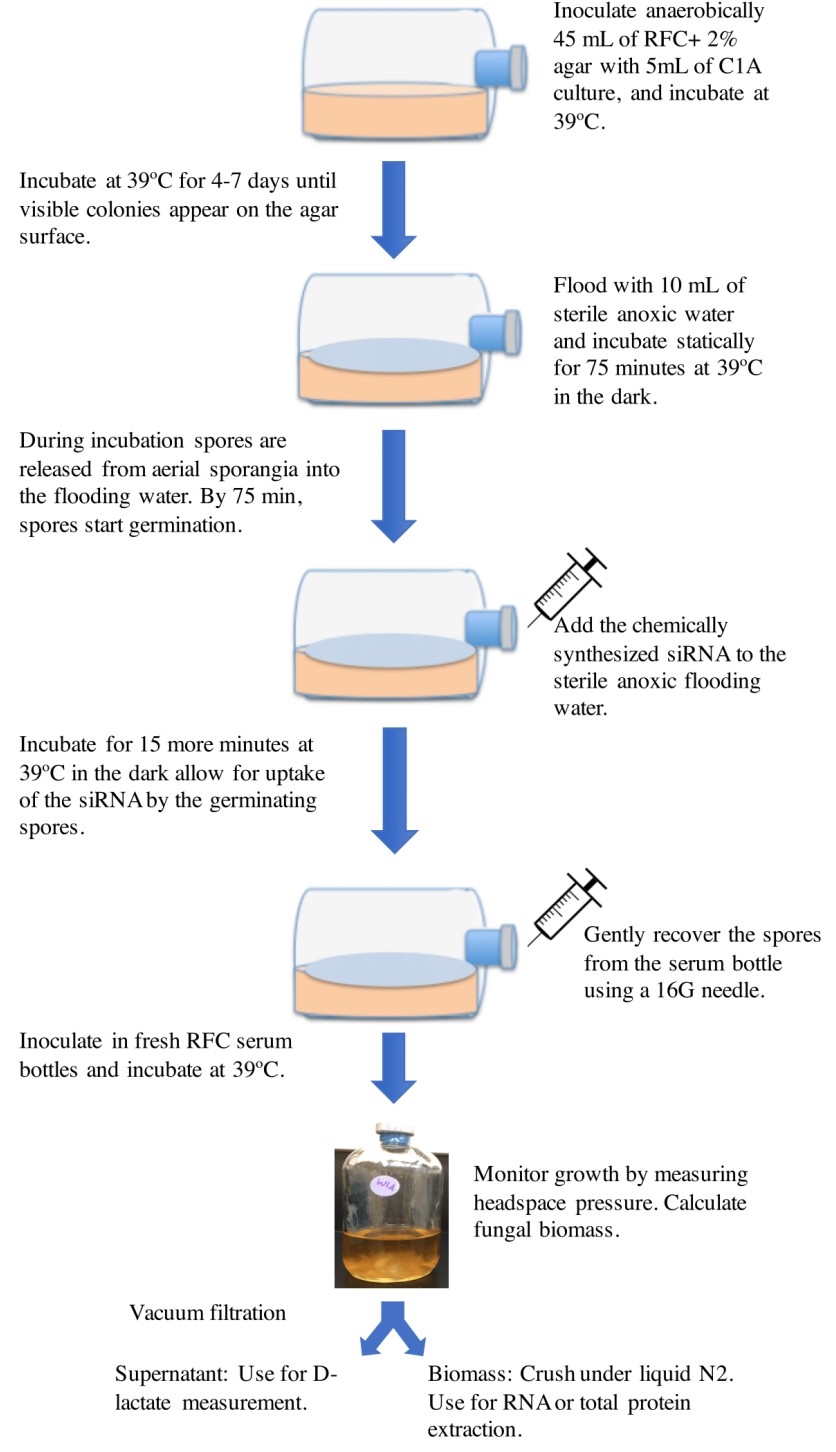

Inoculate anaerobically 45 mL of RFC+ 2% agar with 5mL of C1A culture, and incubate at 39ºC.

Incubate at 39ºC for 4-7 days until visible colonies appear on the agar surface.

Flood with 10 mL of sterile anoxic water and incubate statically for 75 minutes at 39ºC in the dark.

During incubation spores are released from aerial sporangia into the flooding water. By 75 min, spores start germination.

Add the chemically synthesized siRNA to the sterile anoxic flooding water.

Incubate for 15 more minutes at 39ºC in the dark allow for uptake of the siRNA by the germinating spores.

Gently recover the spores from the serum bottle using a 16G needle.

Inoculate in fresh RFC serum bottles and incubate at 39ºC.

Monitor growth by measuring headspace pressure. Calculate fungal biomass.

Vacuum filtration

Supernatant: Use for D-lactate measurement.

Biomass: Crush under liquid N2. Use for RNA or total protein extraction.

**Figure 1  A cartoon depicting the RNAi gene knockdown protocol used in this study.**

### RNA extraction, qRT-PCR, and RNA-seq

RNA was extracted following the protocol in Epicentre® MasterPure™ Yeast RNA Purification Kit, with few modifications as detailed previously (*Calkins & Youssef, 2016*). RNA concentrations were measured using the Qubit® RNA HS Assay Kit (Life Technologies®, Carlsbad, CA, USA). Total RNA was utilized for both transcriptional studies using qRT-PCR, as well as for transcriptomic analysis using RNA-seq.

For transcriptional studies, replicate samples were chosen to cover a range of fungal biomass ranging from 6–22 mgs corresponding to various growth stages. Reverse transcription (cDNA synthesis) was performed using the Superscript IV First-Strand Synthesis System kit for RT-PCR (Life Technologies®, Carlsbad, CA, USA), following the manufacturer's protocols. Quantitative reverse transcription PCR (qRT-PCR) was conducted on a MyIQ thermocycler (Bio-Rad Laboratories, Hercules, CA, USA). *ldhD*, as well as the housekeeping gene glyceraldehyde 3-phosphate dehydrogenase (*GAPDH)*, were amplified using primers designed by the OligoPerfect™ Designer tool (Life Technologies, Carlsbad, CA, USA) (*ldhD*-forward primer: AGACCATGGGTGTCATTGGT, *ldhD*-reverse primer TTCATCGGTTAATGGGCAGT; *GAPDH*-forward primer: ATTCCACTCACG-GACGTTTC, *GAPDH*-reverse primer: CTTCTTGGCACCACCCTTTA). The reactions contained 1 μl of C1A cDNA, and 0.5 μM each of the forward and reverse primers. Reactions were heated at 50 °C for 2 min, followed by heating at 95 °C for 8.5 min. This was followed by 50 cycles, with one cycle consisting of 15 s at 95 °C, 60 s at 50 °C, and 30 s at 72 °C. Using the ΔCt method, the number of copies of *ldhD* is reported relative to the number of copies of *GAPDH* used as the normalizing control.

Transcriptomic analysis was used both to evaluate off-target effects of the chemically synthesized *ldhD* siRNA (transcripts that will be down-regulated in siRNA-treated versus untreated cultures), and to examine the effect of *ldhD* knockdown on other NADH-oxidizing mechanisms to compensate for loss of D-LDH as an electron sink in C1A (transcripts that will be up-regulated in siRNA-treated versus untreated cultures). For transcriptomic analysis, RNA from untreated (*Quoc & Nakayashiki, 2015* biological replicates) as well as siRNA-treated (two biological replicates) cultures was sequenced using Illumina-HiSeq. RNA sequencing as well as sequence processing were as described previously (*Couger et al., 2015*). Briefly, de novo assembly of the generated RNA-Seq reads was accomplished using Trinity (*Haas et al., 2013*), and quantitative levels of assembled transcripts were obtained using Bowtie2 (*Langmead & Salzberg, 2012*). Quantitative values in Fragments Per Kilobase of transcripts per Million mapped reads (FPKM) were calculated in RSEM. edgeR (*Robinson, McCarthy & Smyth, 2010*) was used to determine the transcripts that were significantly up- or down-regulated based on the Benjamini–Hochberg adjusted *p*-value (False discovery rate, FDR). We used a threshold of 10% FDR as the cutoff for determining significantly differentially expressed transcripts.

### Total protein extraction and D-Lactate dehydrogenase enzyme assay

For total protein extraction, replicate samples were chosen to cover a range of fungal biomass ranging from 6–22 mgs corresponding to various growth stages. C1A cells crushed in liquid nitrogen were suspended in 0.5 mL of Tris-Gly buffer (3 g Tris base, 14.4 g Glycine,

H$_2$O up to 1L, pH 8.3), and mixed briefly. Cell debris were pelleted by centrifugation (12,500× g for 2 min at 4 °C) and the sample supernatant containing the total protein extract was carefully transferred into a sterile microfuge tube. Protein concentrations were quantified in cellular extracts using Qubit$^{TM}$ Protein assay kit (Life Technologies, Carlsbad, CA, USA). D-LDH enzyme activity was quantified in the cell extracts using the Amplite$^{TM}$ Colorimetric D-Lactate Dehydrogenase Assay Kit (ATT Bioquest$^{®}$, Sunnyvale, CA, USA), following the manufacturer's protocols.

### Statistical analysis

To evaluate the effect of the siRNA treatment on the transcriptional level of *ldhD* relative to the housekeeping gene *gapdh*, the D-LDH specific activity, as well as the total amount of D-lactate in the culture supernatant, Student *t*-tests were conducted to test for the significance of difference between untreated cultures and *ldhD*-siRNA treated cultures, and *p*-values were compared. To evaluate the effect of the concentration of *ldhD*-siRNA exogenously added to C1A cultures on the level of inhibition of *ldhD* (at the RNA (the transcriptional level of *ldhD* relative to the housekeeping gene *gapdh*), protein (D-LDH specific activity), and metabolite (the total amount of D-lactate in the culture supernatant) levels), Student *t*-tests were conducted to test for the significance of difference between samples treated with different concentrations of *ldhD*-siRNA, and *p*-values were compared.

### Nucleotide accession

This Transcriptome Shotgun Assembly project has been deposited at DDBJ/EMBL/Gen-Bank under the accession GFSU00000000. The version described in this paper is the first version, GFSU01000000.

## RESULTS

### RNAi machinery in the Neocallimastigomycota

The four examined Neocallimastigomycota genomes harbored most of the genes constituting the backbone of the RNAi machinery: ribonuclease III dicer, argonaute, QDE3-homolog DNA helicase, and QIP-homolog exonuclease. Phylogenetically, these genes were closely related to representatives from basal fungal lineages (Figs. 2–4). Gene copies in various genomes ranged between 1 to 4 (Figs. 2–4). However, it is notable that all four examined genomes lacked a clear homolog of RNA-dependent RNA polymerase (RdRP) gene. RdRP has been identified in the genomes of diverse organisms including *Caenorhabditis elegans* (*Smardon et al., 2000*), plants, and the majority of examined fungi (*Cogoni & Macino, 1997a*) but is absent in the genomes of vertebrates and flies; in spite of their possession of a robust RNAi machinery that mediates sequence-specific gene silencing in response to exogenously added dsRNAs.

### Uptake of synthetic siRNA by C1A germinating spores and effect on growth

The addition of fluorescently labeled siRNA targeting *ldhD* transcript to C1A spores at the onset of germination followed by a 15-minute incubation at 39 °C resulted in the uptake of the siRNA by the germinating spores as evident by their fluorescence (Fig. S1A).

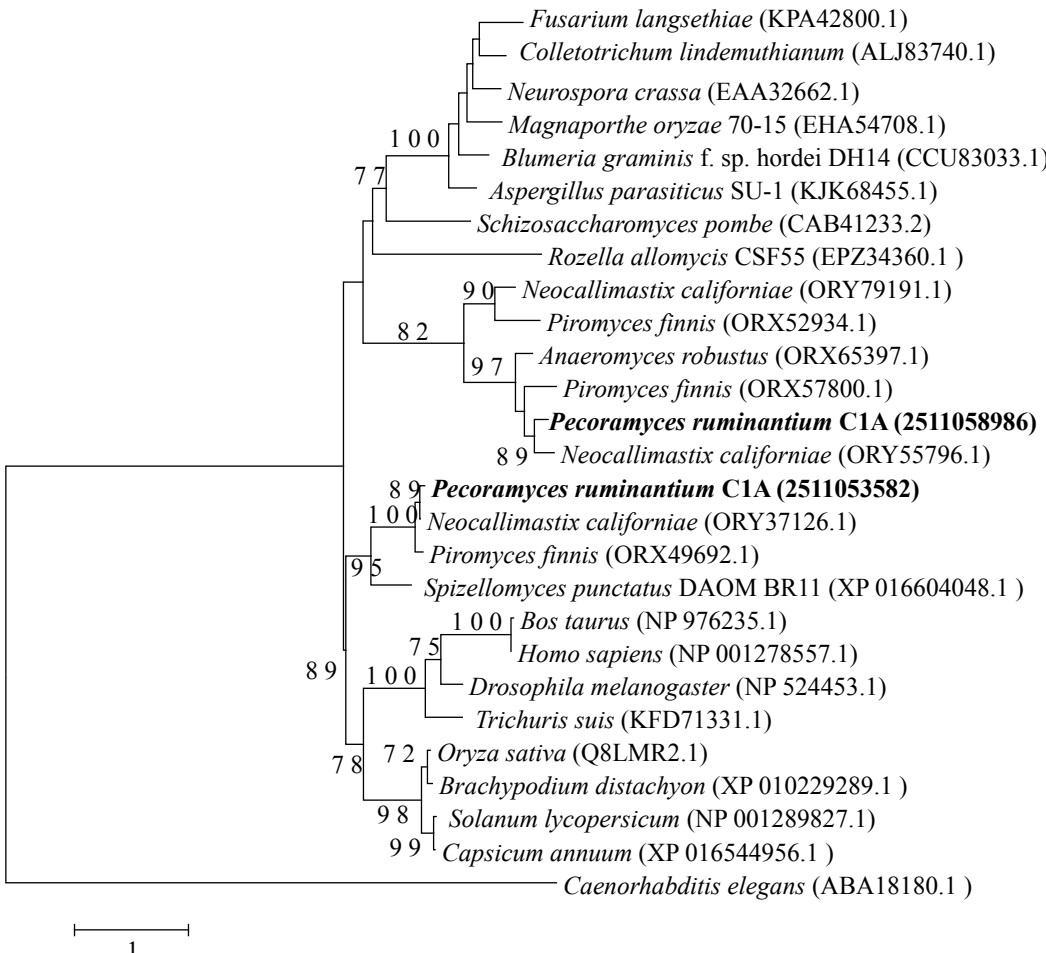

**Figure 2  Neighbor joining phylogenetic tree.** Neighbor joining phylogenetic tree depicting the phylogenetic relationship between *Pecoramyces ruminantium* strain C1A predicted Dicer sequences and those from other fungal and eubaryotic species. Trees were constructed in Mega7 with bootstrap support based on 100 replicates. Bootstrap values are shown for branches with >50 bootstrap support.

Several fields of vision were examined and the number of spores with Cy3-fluoresence, indicative of siRNA uptake, as a percentage of the total number of spores (stained with the nuclear stain DAPI) was evaluated. Under the examined conditions, the majority of the germinating spores picked up the siRNA since 80–90% of spores stained with the nuclear stain DAPI also exhibited Cy3-fluoresence (results from at least four separate experiments). *ldhD*-siRNA-treated spores were collected and used to inoculate fresh RFC liquid media, and the growth rate of these cultures were compared to siRNA-untreated controls. As shown in Fig. S1B, *ldhD*-siRNA treatment had no significant effect on either the rate of fungal growth or the final fungal biomass yield.
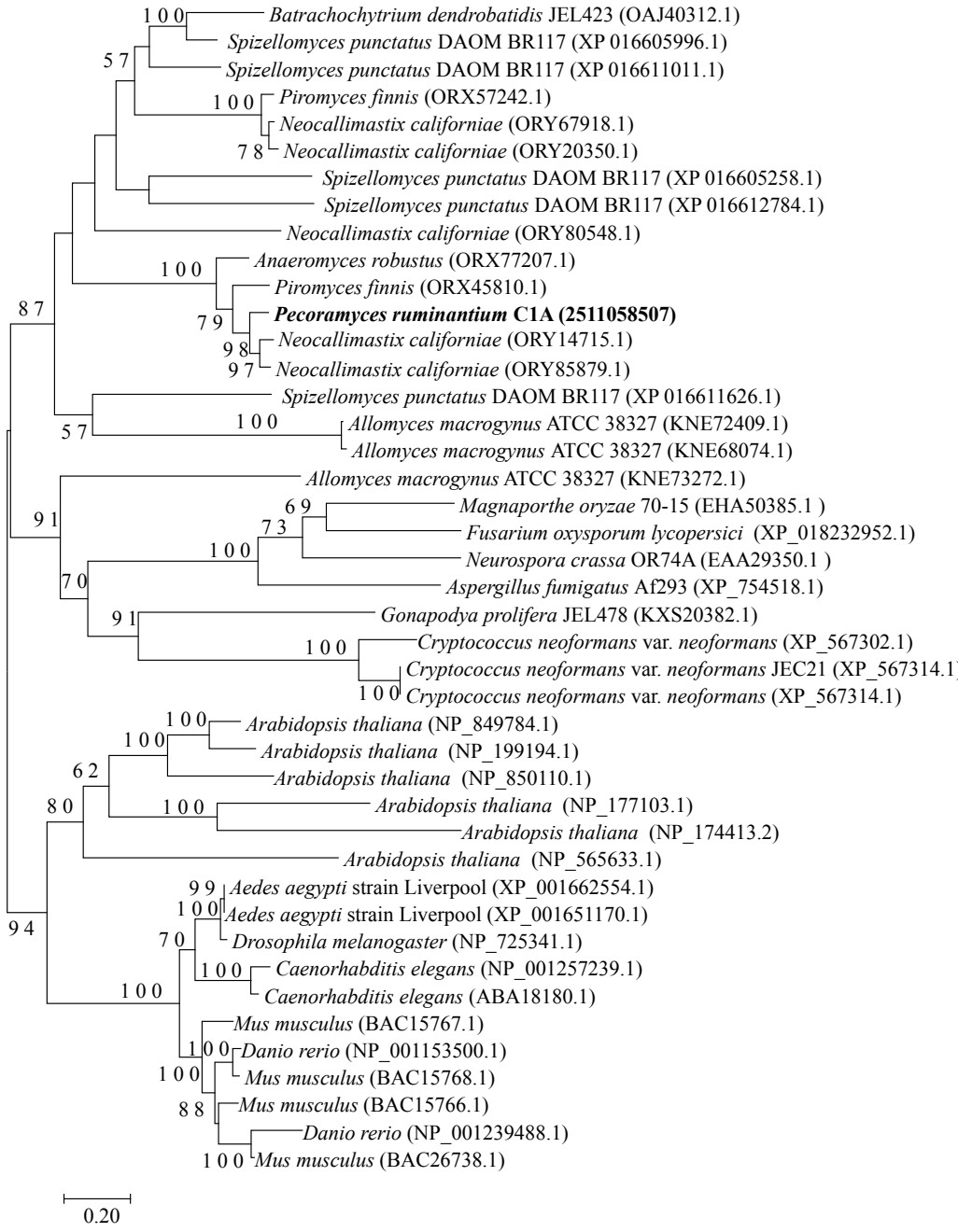

**Figure 3 Neighbor joining phylogenetic tree.** Neighbor joining phylogenetic tree depicting the phylogenetic relationship between *Pecoramyces ruminantium* strain C1A predicted Argonaute sequences and those from other fungal and eukaryotic species. Trees were constructed in Mega7 with bootstrap support based on 100 replicates. Bootstrap values are shown for branches with >50 bootstrap support.

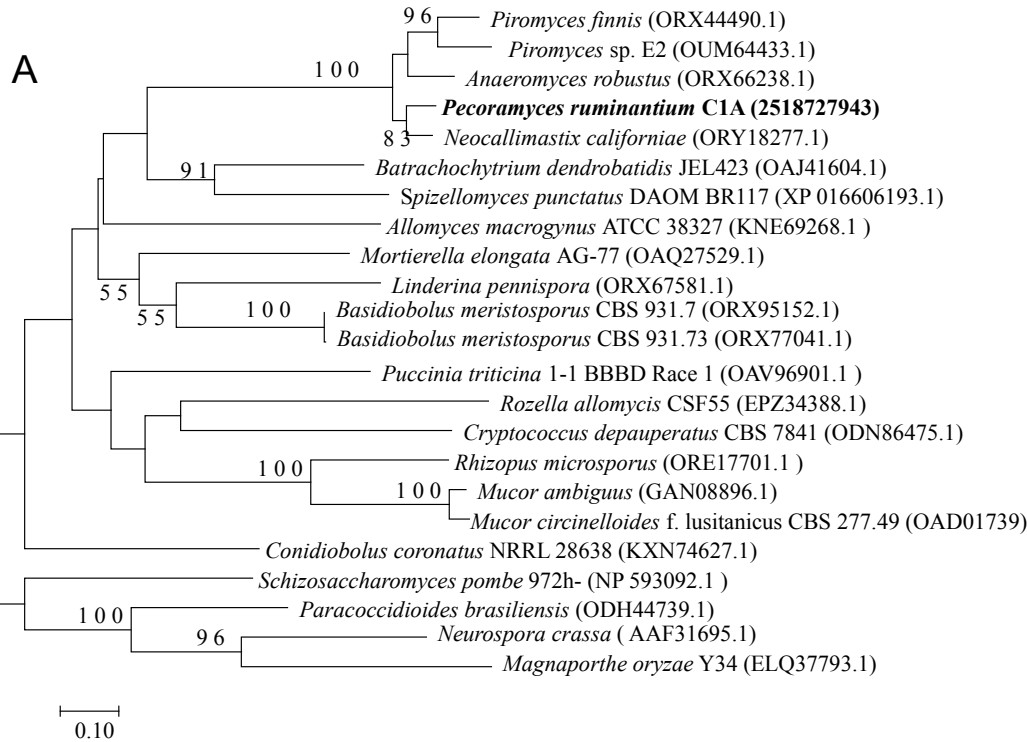

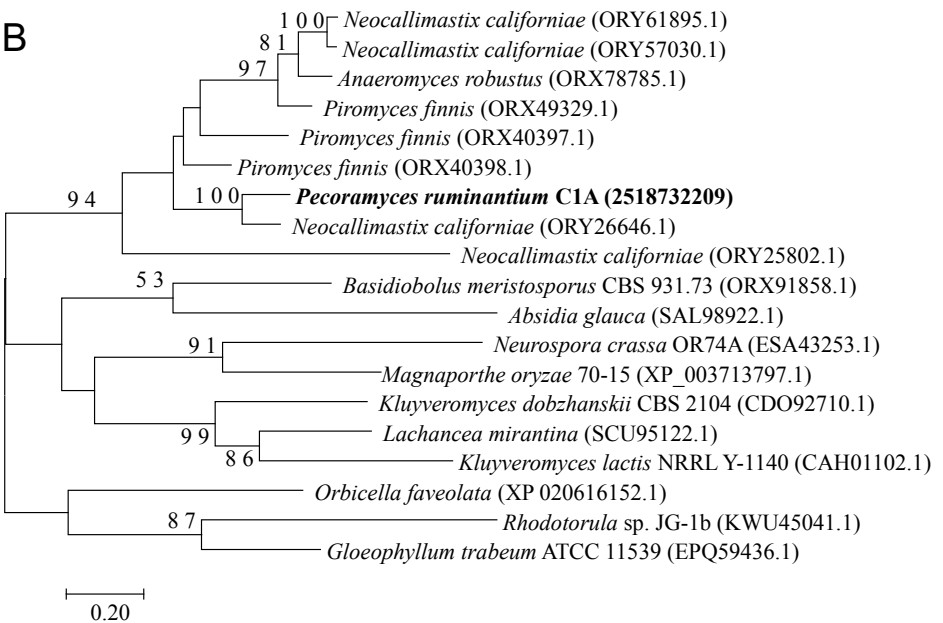

**Figure 4    Neighbor joining phylogenetic tree.** Neighbor joining phylogenetic tree depicting the phylogenetic relationship between *Pecoramyces ruminantium* strain C1A predicted QDE-3 helicase (A) and QIP exonuclease (B) sequences and those from other fungal and eukaryotic species. Trees were constructed in Mega7 with bootstrap support based on 100 replicates. Bootstrap values are shown for branches with >50 bootstrap support.

**Table 1** Effect of the uptake of exogenous *ldhD*-siRNA by C1A germinating spores on the transcriptional level of *ldhD* relative to the housekeeping gene *gapdh*.

| Treatment | Final siRNA conc. (nM) | Copies of *ldhD* relative to *gapdh*[1,2] | Fold change in transcription level ($\Delta\Delta C_t$) compared to untreated samples | Number of biological replicates | Fungal biomass yield (mg) at the time of sacrificing[1] |
|---|---|---|---|---|---|
| | 20 | $4.2E{-}03 \pm 3E{-}03$***,[a] | 0.02 | 4 | $12.3 \pm 5$ |
| *ldhD*-siRNA | 50 | $4.4E{-}03 \pm 2E{-}03$***,[b] | 0.02 | 5 | $9.3 \pm 5.2$ |
| | 75 | $3.6E{-}04 \pm 1.8E{-}04$***,[ab] | 0.0017 | 4 | $15.4 \pm 3.7$ |
| | 100 | $6.1E{-}05 \pm 2.4E{-}05$***,[ab] | 0.0003 | 4 | $15.9 \pm 6$ |
| Untreated | NA | $0.21 \pm 0.04$ | 1 | 5 | $9.6 \pm 2$ |
| Unrelated-siRNA | 50 | $0.26 \pm 0.07$[NS] | 1.29 | 2 | $13.5 \pm 3.8$ |

**Notes.**

[1] Values are average ± standard deviation. Student $t$-test was used to test the significance of the difference of the siRNA-treated samples averages from that of the untreated sample. All *ldhD*-siRNA treated samples showed a significant decrease in the transcriptional level of *ldhD* relative to the housekeeping gene *gapdh* compared to the untreated control. *P*-value of the significant difference; ***, *p*-value <0.00002; **, *p*-value = 0.0012; NS, not significant.

[2,a,b] When comparing the transcriptional level of *ldhD* relative to the housekeeping gene *gapdh* in samples treated with different concentration of *ldhD*-siRNA, a significant difference was observed with concentrations higher than 50 nM (*p*-value <0.05). Samples with the same letter were significantly different.

## Knockdown of *ldhD*-gene by exogenously added *ldhD*-siRNA

### Inhibition at the mRNA level

Table 1 shows the effect of adding exogenous *ldhD*-siRNA on *ldhD* transcriptional level relative to the housekeeping gene glyceraldehyde-3-phosphate dehydrogenase. Results from qRT-PCR revealed that there was an observable decrease in *ldhD* transcription levels in samples treated with *ldhD*-specific siRNA compared to siRNA-untreated samples or unrelated siRNA-treated samples. This effect was significant (Student $t$-test $p$-values $\leq 0.0012$). The inhibitory effect increased with the concentration of *ldhD*-specific siRNA added, and this observed increase in inhibitory effect was significant at siRNA concentrations higher than 50 nM (Student $t$-test $p$-values <0.05). The highest level of inhibition was obtained when 100 nM of the *ldhD*-siRNA was exogenously added to C1A germinating spores, where a four-fold decrease in transcription was observed. C1A cultures treated with the unrelated siRNA showed no significant difference in the transcriptional level of *ldhD* when compared to siRNA-untreated cultures.

### Inhibition at the protein level

Similar to the effect of treatment on the mRNA level, *ldhD*-siRNA-treated samples exhibited a marked decrease in the specific D-LDH activity (Table 2). This effect was significant (Student $t$-test $p$-value $<2 \times 10^{-8}$). The decrease in D-LDH specific activity was dependent on the concentration of siRNA added and ranged from 71–84% reduction compared to siRNA-untreated samples. The highest level of inhibition was obtained when 100 nM of the *ldhD*-siRNA was exogenously added to C1A germinating spores, where an 84% decrease in D-LDH specific activity was observed. When comparing samples treated with different concentration of *ldhD*-siRNA to one another, D-LDH specific activities in samples treated with 20 nM *ldhD*-siRNA were not significantly different from those in samples treated with 50 nM *ldhD*-siRNA ($p > 0.05$). Similarly, D-LDH specific activities in samples treated with 75 nM *ldhD*-siRNA were not significantly different from those in samples treated with

**Table 2  Effect of the uptake of *ldhD*-siRNA by C1A germinating spores on the D-LDH specific activity.**

| Treatment | siRNA concentration (nM) | D-LDH specific activity (U/mg protein)[1] | Fold change in D-LDH specific activity compared to untreated samples | Total number of biological replicates | Fungal biomass yield (mg) at the time of sacrificing[1] |
|---|---|---|---|---|---|
| *ldhD*-siRNA | 20 | $332.2 \pm 90^{***}$ | 0.29 | 6 | $16.5 \pm 5.8$ |
| | 50 | $331.9 \pm 144.5^{***}$ | 0.29 | 17 | $10 \pm 4.3$ |
| | 75 | $194.2 \pm 79^{***}$ | 0.17 | 6 | $12.8 \pm 5.3$ |
| | 100 | $180.6 \pm 131^{***}$ | 0.16 | 6 | $12.7 \pm 7.4$ |
| Untreated | NA | $1157.6 \pm 308.6$ | 1 | 13 | $10.9 \pm 2.9$ |
| Unrelated-siRNA | 50 | $926.4 \pm 69^{NS}$ | 0.8 | 2 | $13.5 \pm 3.8$ |

**Notes.**

[1]Values are average $\pm$ standard deviation. Student $t$-test was used to test the significance of the difference of the siRNA-treated samples averages from that of the untreated sample. All *ldhD*-siRNA treated samples showed a significant decrease in the D-LDH specific activity compared to the untreated control. $P$-value of the significant difference; ***, $p$-value $<2 \times 10^{-8}$; NS, not significant. When comparing samples treated with different concentration of *ldhD*-siRNA to one another, D-LDH specific activities in samples treated with 20 nM *ldhD*-siRNA were not significantly different from those in samples treated with 50 nM *ldhD*-siRNA ($p > 0.05$). Similarly, D-LDH specific activities in samples treated with 75 nM *ldhD*-siRNA were not significantly different from those in samples treated with 100 nM *ldhD*-siRNA ($p > 0.05$). However, D-LDH specific activities in samples treated with 20 nM *ldhD*-siRNA were significantly different from those in samples treated with 75 nM or 100 nM *ldhD*-siRNA, and similarly, samples treated with 50 nM *ldhD*-siRNA were significantly different from those in samples treated with 75 nM or 100 nM *ldhD*-siRNA ($p < 0.05$).

100 nM *ldhD*-siRNA ($p > 0.05$). However, D-LDH specific activities in samples treated with 20 nM *ldhD*-siRNA were significantly different from those in samples treated with 75 nM or 100 nM *ldhD*-siRNA, and similarly, samples treated with 50 nM *ldhD*-siRNA were significantly different from those in samples treated with 75 nM or 100 nM *ldhD*-siRNA ($p < 0.05$). C1A cultures treated with the unrelated siRNA showed no significant difference in D-LDH specific activities when compared to siRNA-untreated cultures (Table 2).

### Effect of ldhD gene knockdown on the extracellular levels of D-lactate in culture supernatants

D-lactate production in C1A culture supernatant is non-linear, with higher amounts of D-lactate produced at later stages of growth (Fig. 5A). D-lactate production in *ldhD*-siRNA-treated cultures was invariably significantly lower when compared to controls (Student $t$-test $p$-value $<0.05$) (Fig. 5B), with the difference especially pronounced at later stages of growth. The level of reduction was dependent on the siRNA concentration added and ranged from 42–86% in the early log phase, 49–67% in the mid log phase, and 57–86% in the late log-early stationary growth phase (Fig. 5B, and Table S1).

### Transcriptomic analysis

Differential gene expression patterns between *ldhD*-siRNA-treated and siRNA-untreated samples were analyzed to identify possible off-target effects of siRNA treatment, i.e., transcripts that were significantly down-regulated in the siRNA-treated cultures. Only 29 transcripts were significantly (FDR $<0.1$) down-regulated (Fig. 6). Predicted functions of these transcripts are shown in Table S2 and included hypothetical proteins ($n = 11$), several glycosyl hydrolases ($n = 5$), and other non-fermentation related functions. Comparison of the siRNA sequence to these 29 transcripts revealed matches to the first seven bases of the *ldhD*-siRNA sequence to only three of the down-regulated transcripts indicating that the off-target effect was mainly not sequence-specific.

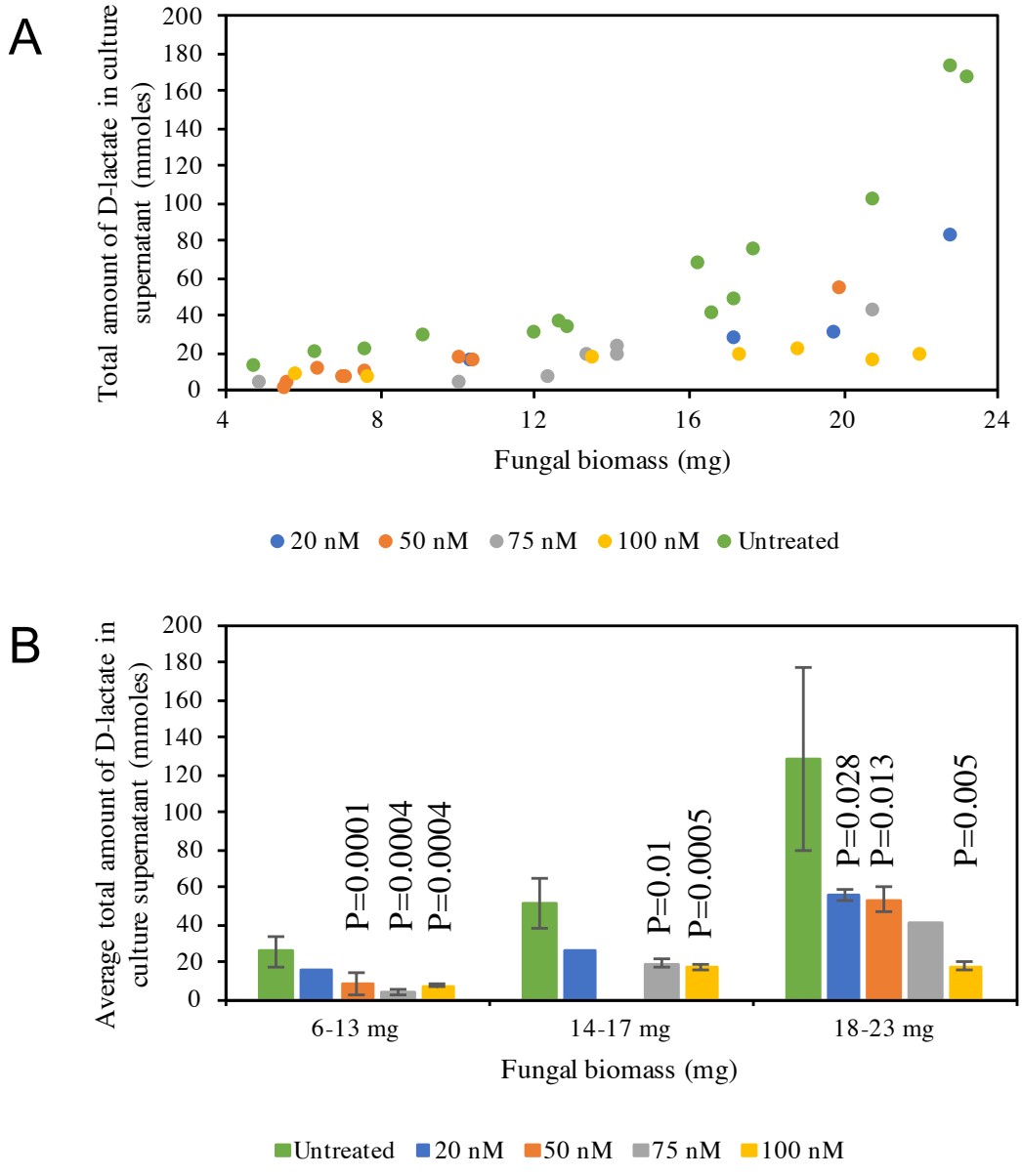

**Figure 5  D-lactate in culture supernatant.** (A) Pattern of D-lactate production in C1A culture supernatant as a factor of fungal biomass. The majority of the D-lactate production occurs at the late log-early stationary phase. Data is shown for both siRNA-untreated cultures (green), as well as *ldhD*-specific siRNA-treated cultures with final concentration 20 nM (dark blue), 50 nM (orange), 75 nM (grey), and 100 nM (yellow). (B) A bar-chart depicting average ± standard deviation (from at least two replicates) of D-lactate levels in C1A culture supernatant during early log (6–13 mg biomass), mid-log (14–17 mg biomass), and late log/early stationary (18–23 mg biomass) phases. Data is shown for both siRNA-untreated cultures (green), as well as *ldhD*-specific siRNA-treated cultures with final concentration 20 nM (dark blue), 50 nM (orange), 75 nM (grey), and 100 nM (yellow).

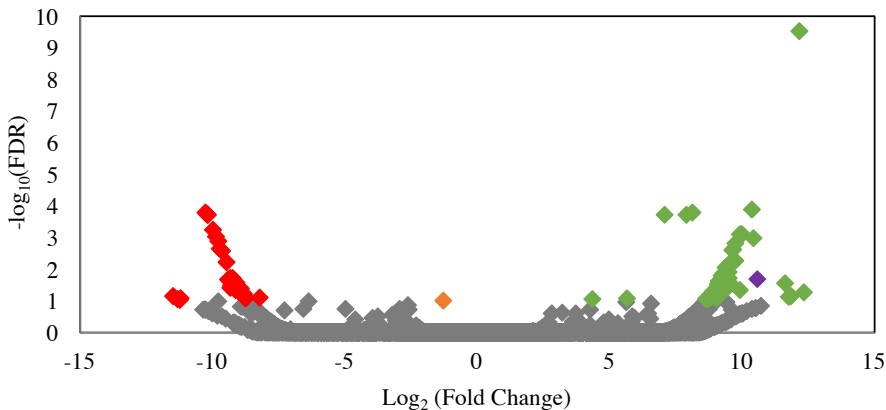

**Figure 6  Volcano plot of the distribution of gene expression for C1A cultures when treated with**
***ldhD*-specific siRNA (50 nM) versus untreated cultures.** The fold change ($\log_2$ (average FPKM in
siRNA-treated cultures/average FPKM in control cultures)) is shown on the $X$-axis, while the significance
of the change ($-\log_{10}$ (false discovery rate)) is shown on the $Y$-axis. Red data points are those transcripts
that were significantly down-regulated ($n = 29$), while green data points are those transcripts that were
significantly up-regulated ($n = 53$). The corresponding IMG gene accession numbers and the predicted
functions for these genes are shown in Table S1. The orange data point corresponds to the D-lactate
dehydrogenase transcript (targeted in the RNAi experiment) with 2.5-fold decrease in FPKM compared
to the untreated control, while the purple data point corresponds to the NAD-dependent 2-hydroxyacid
dehydrogenase (Pfam 00389) transcript (possibly acting to compensate for the loss of NADH oxidation
that occurred as a result of *ldhD* knockdown) with 1,542-fold increase in FPKM compared to the
untreated control.

In an attempt to decipher the impact of inhibiting the D-lactate dehydrogenase enzyme
(one of the major electron sinks in C1A) on the glycolytic and fermentation pathways
in C1A, we investigated the significantly up-regulated transcripts in the siRNA-treated
cultures. A total of 53 transcripts were significantly upregulated in the siRNA-treated
cultures (FDR < 0.1) (Fig. 6). Predicted functions of these transcripts are shown in Table
S2. One transcript encoding NAD-dependent 2-hydroxyacid dehydrogenase (Pfam 00389)
was significantly upregulated (1,542-fold) in the siRNA-treated cultures ($P$-value = 0.02).
Enzymes belonging to this family act specifically on the D-isomer of their substrates
(*Dengler et al., 1997*). In case of D-LDH inhibition in the siRNA-treated cultures, the Pfam
00389 enzyme might act to compensate for the loss of NADH oxidation by acting on an
alternate substrate (e.g., hydroxypyruvate, 2-oxoisocaproate, or other 2-oxo carboxylic
acids) and reducing it as a sink of electrons to regenerate NAD. However, it is difficult to
know the actual substrate based on sequence data alone. Transcripts of other glycolytic and
fermentative enzymes of C1A were not differentially expressed in siRNA-treated cultures
(Table S2).

## DISCUSSION

Here, we explored the feasibility of RNA interference for targeted gene silencing in the
anaerobic gut fungi (phylum Neocallimastigomycota) via the exogenous addition of
synthetic double stranded siRNAs targeting the *ldhD* gene to *Pecoramyces ruminantium*

strain C1A germinating spores. We show that ds-siRNA was uptaken by germinating spores, and, as a consequence, the transcription of the target gene (*ldhD*) was down-regulated (Table 1), leading to lower D-LDH enzymatic activity (Table 2) and lower D-lactate concentration in the culture supernatant (Fig. 5).

In general, the fungal RNAi machinery encompasses Dicer (Dic) enzyme(s), Argonaute (Ago) protein(s), RNA-dependent RNA polymerase (RdRP) enzyme, QDE3-like DNA helicase, and Argonaute-interacting exonuclease (QIP-like). Genomes of Neocallimastigomycota representatives belonging to four genera (*Pecoramyces*, *Neocallimastix*, *Piromyces*, and *Anaeromyces*) encode at least one copy of Dic, Ago, QDE3-like helicase, and QIP exonuclease. However, all genomes lacked a clear homolog of RdRP. The absence of an RdRP homolog is not uncommon. While present in almost all studied fungi, RdRP seems to be missing from the genomes of other basal fungal phyla (Chytridiomycota and Blastocladiomycota) representatives (*Choi et al., 2014*; *Farrer et al., 2017*). The absence of clear RdRP homologues in the Neocallimastigomycota and related basal fungal phyla despite their presence in other fungi could suggest that either an RdRP is not involved in dsRNA-mediated mRNA silencing as shown before in mammals (*Stein et al., 2003*). Alternatively, RNA-dependent RNA polymerase activity could be mediated through a non-canonical RdRP in basal fungi, e.g., the RNA polymerase II core elongator complex subunit Elp1 shown to have RdRP activity in *Drosophila*, as well as *Caenorhabditis elegans*, *Schizosaccharomyces pombe*, and human (*Birchler, 2009*; *Lipardi & Paterson, 2009*).

We chose as a gene knockdown target the D-Lactate dehydrogenase gene (*ldhD*) that mediates NADH-dependent pyruvate reduction to D-lactate, for several reasons. First, the gene is present as a single copy in the genome. Second, quantification of the impact of *ldhD* gene knockdown is readily achievable in liquid media at the RNA (using RT-PCR and transcriptomics), and protein (using specific enzyme activity assays) levels, as well as phenotypically (by measuring D-lactate accumulation in the culture media); providing multiple lines of evidence for the efficacy of the process. Finally, D-lactate dehydrogenase is part of the complex mixed acid fermentation pathway in *P. ruminantium* (*Ranganathan et al., 2017*; *Youssef et al., 2013*) and other anaerobic gut fungi, and we sought to determine how blocking one route of electron disposal could lead to changes in C1A fermentation end products.

*ldhD*-siRNA-treated cultures showed a significant reduction in *ldhD* gene transcription and D-LDH enzyme activity. Both of these effects were dependent on the concentration of siRNA added (Tables 1 and 2) similar to previous reports in filamentous fungi (*Eslami et al., 2014*; *Jöchl et al., 2009*; *Kalleda, Naorem & Manchikatla, 2013*; *Mousavi et al., 2015*). We show that the addition of 100 nM of *ldhD*-siRNA resulted in a four-fold reduction in *ldhD* transcription, 84% reduction in D-LDH specific activity, and 86% reduction in D-lactate concentration in culture supernatant. The fact that targeted gene silencing using exogenously added gene-specific siRNA results in reducing rather than completely abolishing gene function is an important advantage of RNAi approaches allowing functional studies of housekeeping or survival-essential genes.

While initial studies of gene silencing using exogenously added siRNAs suggested that the process was highly sequence-specific (*Elbashir et al., 2001*; *Tuschl et al., 1999*), subsequent

studies showed silencing of off-target genes based on less than perfect complementarity between the siRNA and the off-target gene (*Jackson et al., 2003*). Here, we used RNA-seq to quantify the off-target effects of *ldhD*-siRNA. In contrast to previous studies that used similar approaches to quantify RNAi off-targets (*Li-Byarlay et al., 2013*), we show here that the off-target effects of *ldhD* silencing were minimal (only 29 transcripts out of 55,167 total transcripts were differentially down-regulated as a result of siRNA treatment) and appeared to be not sequence-specific.

Currently, and due to their strict anaerobic nature, there are no established procedures for genetic manipulations (e.g., gene silencing, insertion, deletion, and mutation) of AGF leading to a paucity of molecular biological studies of the phylum. This is in stark contrast to the rich body of knowledge available on genetic manipulations of various aerobic fungal lineages (*Eslami et al., 2014*; *Kalleda, Naorem & Manchikatla, 2013*; *Mousavi et al., 2015*; *Khatri & Rajam, 2007*; *Michielse et al., 2008*; *Minz & Sharon, 2010*). Our work here represents a proof of principal of the feasibility of the RNAi approach in AGF, and opens the door for genetic manipulation and gene function studies in this important group of fungi.

## CONCLUSIONS

Anaerobic gut fungi (AGF) have a restricted habitat in the herbivorous gut. Due to their anaerobic nature, gene manipulation studies are limited hindering gene-targeted molecular biological manipulations. We used an AGF representative, *Pecoramyces ruminantium* strain C1A, to study the feasibility of using RNA interference (RNAi) for targeted gene silencing. Using D-lactate dehydrogenase (*ldhD*) gene as a target, we show that RNAi is feasible in AGF as evidenced by significantly lower gene transcriptional levels, a marked reduction in encoded enzymatic activity in intracellular protein extracts, and a reduction in D-lactate levels accumulating in the culture supernatant. To our knowledge, this is the first attempt of gene manipulation studies in the AGF lineage and should open the door for gene silencing-based studies in this fungal clade.

### Funding

This work was supported by the National Science Foundation Grant award number 1557102. The funders had no role in study design, data collection and analysis, decision to publish, or preparation of the manuscript.

### Grant Disclosures

The following grant information was disclosed by the authors:
National Science Foundation: 1557102.

### Competing Interests

The authors declare there are no competing interests.

## Author Contributions

- Shelby S. Calkins performed the experiments, analyzed the data, wrote the paper, prepared figures and/or tables, reviewed drafts of the paper.
- Nicole C. Elledge performed the experiments, reviewed drafts of the paper.
- Katherine E. Mueller performed the experiments, analyzed the data, reviewed drafts of the paper.
- Stephen M. Marek reviewed drafts of the paper.
- MB Couger performed the experiments, analyzed the data, contributed reagents/materials/analysis tools, reviewed drafts of the paper.
- Mostafa S. Elshahed analyzed the data, contributed reagents/materials/analysis tools, wrote the paper, prepared figures and/or tables, reviewed drafts of the paper.
- Noha H. Youssef conceived and designed the experiments, analyzed the data, contributed reagents/materials/analysis tools, wrote the paper, prepared figures and/or tables, reviewed drafts of the paper.

## Data Availability

This Transcriptome Shotgun Assembly project has been deposited at DDBJ/EMBL/-GenBank under the accession GFSU00000000. The version described in this paper is the first version, GFSU01000000.

## Supplemental Information

Supplemental information for this article can be found online at http://dx.doi.org/10.7717/peerj.4276#supplemental-information.

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
