# Peer review of "Development of an RNA interference (RNAi) gene knockdown protocol in the anaerobic gut fungus Pecoramyces ruminantium strain C1A"

_PeerJ, doi:10.7717/peerj.4276_

## Round 0.1 · original submission · Minor Revisions

· Academic Editor

Minor Revisions

Please pay special attention to the comments of Reviewer 1 regarding the need for statistical evaluation of the results in Tables 1 and 2. While we do not require additional experiments, the questions raised by the reviewers need to be addressed and the manuscript modified accordingly.

·

Basic reporting

The manuscript is clearly written in the professional language and article structure with sufficient supporting tables and figures. The manuscript showed relevant results to answer hypothesis in RNAi Pecoramyces ruminantium strain C1A.

Experimental design

Generally, this primary research tried to fill in the knowledge gap on the development of RNAi in Pecoramyces ruminantium strain C1A that has not been demonstrated in anaerobic fungi. Although more rigorous investigation must be required to prove the deep mechanism of RNAi in Pecoramyces ruminantium, the methods described in this study is sufficient to know if siRNA delivery can work well in Pecoramyces ruminantium.

However, there is the lack of statistical approach to indicate the efficient and inefficient difference in table 1 and table 2. In material and methods, the authors should compare genes encoding Dicer, Ago, QIP, QDE3, RdRp in Pecoramyces ruminantium with those that were reported in filamentous fungi such as N.crassa, M.oryzae, etc.

Validity of the findings

1. RNAi machinery in the Neocallimastigomycota
Although the authors have identified at least 2 Dicer protein, one protein for each Ago, QDE2, QIP protein and no protein homologs of RdRp in Pecoramyces ruminantium, It is not clear that which Dicer protein involves in RNAi machinery as described in filamentous fungi. In addition, the phylogenetic relationship should be shown between those proteins and RNAi components in Ascomycete group.

2. In the figure S1A, the author should explain how to evaluate the rates of fungal growth and fungal biomass. They should add more result on the rate of fungal growth in this case.

3. In the line 250-251, the author mentioned the rates of the germinating spores that picked up the siRNA, were 80-90%. Do you have any evidence supporting this conclusion because I think that those rates play important role in silencing efficiency? Are you sure that all experiments showed the same percentage if siRNA intake?

4. In table 1, could you please explain why the fungal biomass yield treated with 150 nM siRNA was lower than that with 100 nM siRNA that was considered as the most silencing efficiency (4 folds decrease of ldhD transcriptional level)?

5. I found that the results of 100-150 nm siRNA treatments were not identical in table 1 and table 2. Herein, silencing efficiency of LDHD gene was observed at 100 nM siRNA with 4 folds decrease of ldhD transcriptional level compared with untreated control and unrelated siRNA (table ). However, I found that D-LDH specific activity when treated with 150 nM siRNA, was lowest resulting the lowest fungal biomass yield in comparison with the treatment of 100 nM siRNA. Please explain clearly about this in the discussion part.

6. In figure 3, the authors should explain why they do not evaluate the correlations between time intervals of fungal culturing and silencing efficiency of the target gene. Due to artificial siRNA mediated gene silencing, the efficiency could be decreased by time.

Additional comments

Generally, the authors have tried to prove that RNAi could work in the anaerobic gut fungus, Pecoramyces ruminantium and provide RNA gene knockdown protocol for this purpose. I think the author should make clear some comments in this manuscript that should be improved upon before acceptance.

Reviewer 2 ·

Basic reporting

The manuscript is very well-written, clear, and concise. Appropriate references are included and the figures are generally acceptable. One suggestion would be to include a higher magnification of the spore siRNA uptake panels in Figure S1, and the inclusion of perhaps an outline of an individual spore at higher mag to give an clear indication of the boundary the is encompassing the fluorescence signal.

Experimental design

The experimental design is made clear, and appropriate controls are included. The methods are described clearly and a strong rationale for the experimental plan is presented. The RNA-seq data presented demonstrate some potential off-target or secondary effects. Are there any processes or functions that are enriched in the RNA-seq data that could tell you if these are related processes impacted by the siRNA or if they are truly random/off target effects? GO analysis perhaps?

Validity of the findings

The authors demonstrate clearly by a number of corroborating experiments that the siRNA is indeed being taken into the spores and is exerting a repressive effect on the target mRNA and the activity encoded by the target mRNA. Taken together with the RNA-seq data, there is compelling evidence that RNAi-mediated silencing is functional and may be a useful tool in expoloring further the biology of Pecoramyces ruminantium. Appropriate statistics are provided, and the data are presented both in such a way that any manipulation or normalization is apparent.

Additional comments

Summary of minor suggestions from above:
1. Include a higher magnification of the spore siRNA uptake panels in Figure S1, and the inclusion of perhaps an outline of an individual spore at higher mag to give an clear indication of the boundary the is encompassing the fluorescence signal.
2. Are there any processes or functions that are enriched in the RNA-seq data that could tell you if these are related processes impacted by the siRNA or if they are truly random/off target effects? GO analysis perhaps?

---

## Round 0.2 · accepted · Accept

· Academic Editor

Accept

I apologize for the prolonged turn-around time, which resulted from a misunderstanding in the decision process.